# CASCADE STYLE TRANSFER

## ABSTRACT

Recent studies have made tremendous progress in style transfer for specific domains, e.g., artistic, semantic and photo-realistic. However, existing approaches have limited flexibility in extending to other domains, as different style representations are often specific to particular domains. This also limits the stylistic quality. To address these limitations, we propose Cascade Style Transfer, a simple yet effective framework that can improve the quality and flexibility of style transfer by combining multiple existing approaches directly. Our cascade framework contains two architectures, i.e., Serial Style Transfer (SST) and Parallel Style Transfer (PST). The SST takes the stylized output of one method as the input content of the others. This could help improve the stylistic quality. The PST uses a shared backbone and a loss module to optimize the loss functions of different methods in parallel. This could help improve the quality and flexibility, and guide us to find domain-independent approaches. Our experiments are conducted on three major style transfer domains: artistic, semantic and photo-realistic. In all these domains, our methods have shown superiority over the state-of-the-art methods.

## 1 INTRODUCTION

Given the content and style images, the goal of style transfer is to synthesize an image that preserves some notion of the content but carries characteristics of the style. Recently, the seminal work of Gatys et al. (2015) firstly captured the style of artistic images and transferred it to other images using Convolutional Neural Networks (CNNs). Since then, various Neural Style Transfer (NST) Jing et al. (2017) methods have been advanced and obtained visually pleasing results.

Despite the recent rapid progress, these existing works often limited to one or few specific domains (in this paper, we mainly focus on three domains: artistic, semantic and photo-realistic). For instance, Li et al. (2017b); Huang & Belongie (2017); Gatys et al. (2016) can transfer the artistic styles well, but they perform poorly on the style transfer of photographs and corresponding semantics. Luan et al. (2017); Li et al. (2018) specialize in photo-realistic style transfer, and Li & Wand (2016); Champandard (2016) mainly target semantic style transfer. Fortunately, there are some multi-domain approaches which can perform well on multiple domains, e.g., Liao et al. (2017) can perform well on semantic and photo-realistic style transfer, Gu et al. (2018) is suitable for artistic and semantic style transfer, and Li et al. (2019) can adapt to artistic and photo-realistic style transfer. Nevertheless, they still have some limitations, and the quality could be further improved (see Fig. 1 (a)). That is to say, nowadays, it is still inconvenient to users that have to choose the appropriate methods for specific domains. In this sense, finding a common approach which could perform well in all style transfer domains is extremely hard but significant.

As a coin has two sides, every existing NST method has both advantages and shortcomings. Fig. 1 (b) shows some typical examples, we can observe that using Gram matrices Gatys et al. (2016) to transfer the artistic styles performs well on global color, but fails to capture enough local patterns (e.g., circles and droplets). Patch-based method Li & Wand (2016) can alleviate this problem, but may cause insufficient color. *Is there a way to combine the advantages of both and overcome their shortcomings?* Obviously, redesigning a new algorithm is difficult, why not use some simpler ways, such as combining existing methods directly through some general architectures?

Based on the above analyses, we propose Cascade Style Transfer (CST) mainly for two targets, i.e., higher quality and higher flexibility, and design two architectures, i.e., the Serial Style Transfer (SST) and the Parallel Style Transfer (PST) for these targets. In this work, we first revisit and

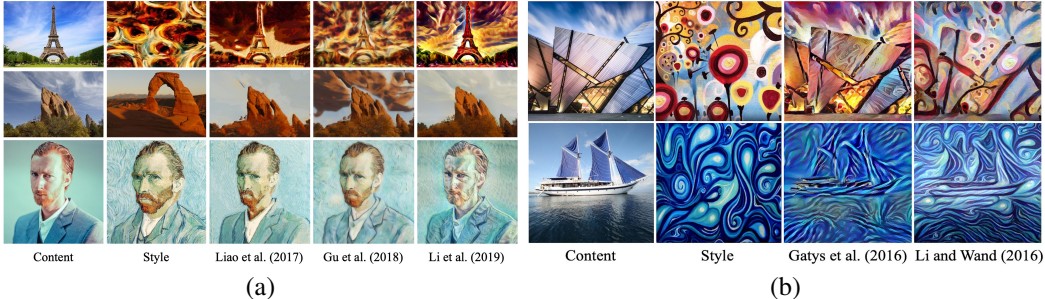

Figure 1: (a) Examples of multi-domain methods. The rows from top to bottom show artistic, photo-realistic and semantic style transfer, respectively. (b) Typical examples of artistic style transfer.

demonstrate the impact of different initialization strategies on style transfer, and inspired by this, design our SST for higher quality domain-specific style transfer. Moreover, we develop upon this and further propose our PST for more flexible style transfer, this could guide us to create domain-independent approaches. As far as we know, this is the first paper to propose domain-independent style transfer (note that this kind of approach is flexible for arbitrary images in *arbitrary* domains, while existing so-called arbitrary style transfer methods are only flexible for arbitrary images in *specific* domains), and also the first attempt to combine multiple existing approaches directly to improve the quality and flexibility of style transfer. The main contributions of our work are:

• We revisit the initialization of style transfer, and demonstrate that initialization can play an important role in improving the quality of style transfer.

• We propose a serial architecture for cascade style transfer, it is simple yet effective, which could help improve the quality of domain-specific style transfer.

• We first propose domain-independent style transfer, and design a parallel architecture to help improve the quality and flexibility of style transfer.

## 2 RELATED WORK

Our cascade style transfer can be related to the most style transfer methods. In this paper, we mainly focus on the NST methods in three major domains: artistic, semantic and photo-realistic.

**Artistic style transfer.** This domain is dedicated to transferring the global artistic styles (e.g., abstract, painterly or sketch). The most representative work is Gatys et al. (2015). This method could produce amazing results but suffers from a slow iterative optimization procedure. To address it, Johnson et al. (2016) and Ulyanov et al. (2016; 2017) trained feed-forward generative networks for fast artistic style transfer. But one limit is that each model is trained to transfer exactly one fixed style. Some methods Dumoulin et al. (2017); Zhang & Dana (2018); Li et al. (2017a); Chen et al. (2017) further incorporated multiple styles into one single model, but they are still limited to a fixed number of pre-trained styles. Recently, several methods Li et al. (2017b); Huang & Belongie (2017); Li et al. (2019) were proposed to allow artistic style transfer for arbitrary images.

**Semantic style transfer.** Transferring the styles between the corresponding semantic regions of the style and content images is referred to as semantic style transfer. The most representative work is Li & Wand (2016). They combined Markov Random Fields (MRFs) and CNNs to match the most similar local neural patches of the style and content images. Later, Champandard (2016) incorporated the segmentation masks for stricter semantic constraints. Recently, Liao et al. (2017) proposed Deep Image Analogy for accurate semantic-level patch match. Gu et al. (2018) used Deep Feature Reshuffle to consider both global and local information. Mechrez et al. (2018) proposed an alternative contextual loss for segmentation-free semantic style transfer. Moreover, some feed-forward methods Chen & Schmidt (2016); Lu et al. (2017); Sheng et al. (2018); Park & Lee (2019); Yao et al. (2019) were also proposed for fast and real-time semantic style transfer.

**Photo-realistic style transfer.** Photo-realistic style transfer seeks to transfer the style of a reference style photo onto other pictures. The greatest characteristic is that both the global structures and

detailed contours in the content images should be preserved during the process. Traditional methods based on Global Reinhard et al. (2001); Pitie et al. (2005) and Local Laffont et al. (2014); Shih et al. (2014) are slow in practice and limited in specific scenarios (e.g., outdoor scenes or headshot portraits). Recently, Luan et al. (2017) incorporated a new loss term to the optimization objective of Gatys et al. (2015) to improve the photorealism of stylization outputs. Li et al. (2018) introduced a closed-form solution consisting of a stylization and a smoothing step for faster speed. More recently, Li et al. (2019) have also shown an effective approach for fast photo-realistic style transfer.

Despite the fact that current NST methods have shown good performance in specific domains, there are few studies on how to combine them directly to improve the quality and flexibility. In our work, we select Gatys et al. (2016); Li et al. (2017b); Huang & Belongie (2017); Li et al. (2019; 2018); Luan et al. (2017); Sheng et al. (2018); Liao et al. (2017); Gu et al. (2018); Li & Wand (2016); Champandard (2016) for our studies mainly because of their representativeness in specific domains.

# 3 REVISIT INITIALIZATION IN STYLE TRANSFER

Initialization is the first yet important step in almost all style transfer algorithms. Although some papers Gatys et al. (2016); Nikulin & Novak (2016) have discussed the impact of the initialization on style transfer, they only use white noise, the content image or the style image, and the evaluation criteria are only based on the qualitative aspect. Here, back to the most original NST algorithms, we revisit and compare more initialization strategies from both qualitative and quantitative aspects.

Our experiments are based on two most original NST methods Gatys et al. (2016); Li & Wand (2016). These methods always initialize with white noise or the content image and iteratively optimize pixels to match the content features of the content image and style features of the style image. Besides white noise, the content and style images, we try and compare four other initialization strategies: salt and pepper noise, poisson noise, the stylized results of other methods (SROOM), and moreover, replacing the content images with SROOM and then initializing with them (RC-SROOM).

**Qualitative Comparisons:** Fig. 2 demonstrates the qualitative comparisons. As we can see, initializing with the content image produces results with insufficient color (column a). Results generated from the style image introduce some undesired color (column b). Using salt and pepper noise produces over-bright results, and using poisson noise produces darker results (column d and e). By contrast, white noise initialization yields satisfying results (column c), but the results of SROOM performs better on overall effect (column f). Remarkably, using RC-SROOM (column g) can dramatically improve the effect and absorb the merits (as shown and discussed in Fig. 1 (b)) of both methods. More results can be found in appendix.

**Quantitative Comparisons:** Fig. 3 shows the quantitative comparisons. We evaluate these initialization strategies on method Gatys et al. (2016), the optimization is conducted by *Adam* method, and stopped at 1000 iterations. We can see that initializing with images (i.e., content, style, SROOM and RC-SROOM) makes loss fall faster, while initializing with noise (i.e., white noise, salt and pepper noise, and poisson noise) decreases loss more steadily. It is worth noting that using RC-SROOM achieves much lower total, content and style loss than all other strategies.

**Conclusion:** In this section, we have demonstrated that initialization plays an important role in improving the quality of style transfer. Compared with other initialization strategies, RC-SROOM has outstanding performance. More importantly, this could help produce higher quality style transfer results that absorb the merits of multiple methods. Inspired by this, we propose cascade style transfer, which will be presented in latter sections, and in turn verify this conclusion.

# 4 CASCADE STYLE TRANSFER

In this paper, we define cascade style transfer as combinations of different NST methods. It contains two architectures: the serial style transfer and the parallel style transfer.

## 4.1 SERIAL STYLE TRANSFER (SST)

As shown in Fig. 4 (a), serial style transfer serially connects multiple style transfer methods. Let $\vec{c}$, $\vec{s}$ and $\vec{x}_i$ be the content image, the style image and the stylized result of method $i$. The style transfer

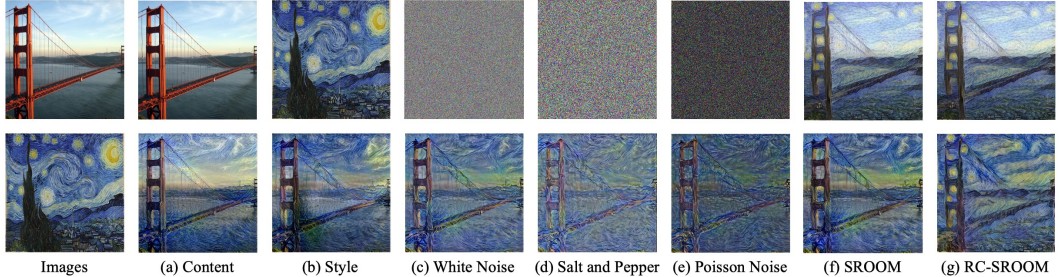

| Images | (a) Content | (b) Style | (c) White Noise | (d) Salt and Pepper | (e) Poisson Noise | (f) SROOM | (g) RC-SROOM |

Figure 2: Qualitative comparisons of different initialization strategies. The first column shows the content image (top) and the style image (bottom). The other columns show the initialization images (top) and the corresponding style transfer results (bottom) of method Gatys et al. (2016). In column (f), we initialize with the default stylized result of method Li & Wand (2016) (SROOM). In column (g), we replace the content image with SROOM and then initialize with it (RC-SROOM).

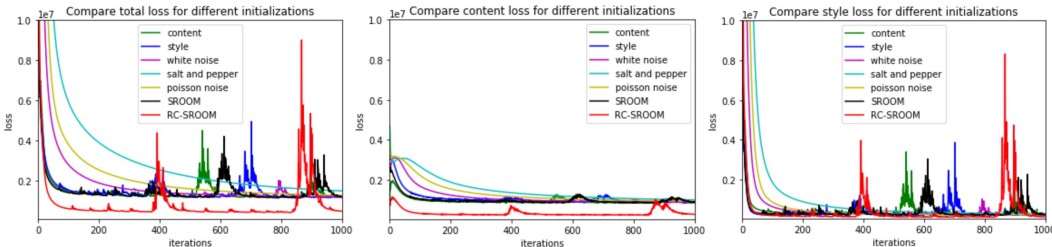

Figure 3: Quantitative comparisons of different initialization strategies on total (left), content (center) and style (right) loss.

process of method $i$ is denoted as $f_i$. Specifically, for the first method, we use $f_1(\vec{c}, \vec{s}, d)$ to denote transferring the style of $\vec{s}$ to $\vec{c}$ by method 1 with the *default* initialization settings. For others, we use $f_i(\vec{x}_{i-1}, \vec{s})$ to denote initializing with $\vec{x}_{i-1}$ and then transferring the style of $\vec{s}$ to $\vec{x}_{i-1}$ by method $i$. Our serial style transfer can be formulated as

$$\vec{x}_i = \begin{cases} f_1(\vec{c}, \vec{s}, d) & \text{if } i = 1 \\ f_i(\vec{x}_{i-1}, \vec{s}) & \text{otherwise.} \end{cases} \tag{1}$$

## 4.2 PARALLEL STYLE TRANSFER (PST)

As far as we know, current NST methods are mainly conducted in two different ways. One is based on VGG Simonyan & Zisserman (2014), iteratively optimizing the pixels of input images. The other is training a feed-forward network to directly generate the stylized results. Here, to demonstrate our PST more intuitively, we design a simple parallel architecture based on the former way.

As shown in Fig. 4 (b), our PST contains two important parts. One is the shared backbone (e.g., VGG-19), it is mainly used for feature extraction and error back-propagation. The other is the loss module, it is used to combine loss functions of different methods.

Let $\mathcal{L}_i$ denote the loss function of method $i$, "$\oplus$" denote a linear combination operation between different loss functions. We give hyperparameter $\omega_i$ to weight every loss function $\mathcal{L}_i$. The total loss is defined as follows:

$$\mathcal{L}_{total}(i) = \begin{cases} \omega_1 \mathcal{L}_1 & \text{if } i = 1 \\ \mathcal{L}_{total}(i-1) \oplus \omega_i \mathcal{L}_i & \text{otherwise.} \end{cases} \tag{2}$$

We compute its gradients with respect to the pixel values and use them to iteratively update the input images. In this way, all methods can be optimized in parallel with the total loss.

To verify the effectiveness of our PST, we design a specific parallel scheme *ParallelNet* based on four popular domain-specific style transfer methods Champandard (2016); Li & Wand (2016); Gatys

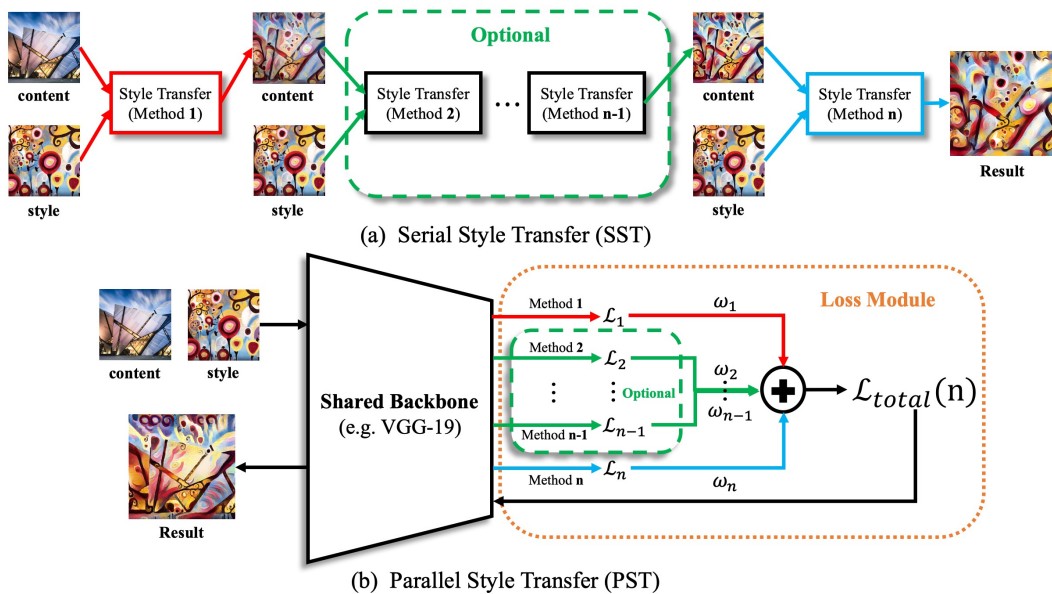

Figure 4: Our proposed cascade style transfer, which contains two architectures: a) serial style transfer, b) parallel style transfer.

et al. (2016); Luan et al. (2017). The principles for selecting the appropriate approaches will be discussed in later sections. The detailed combination procedure is as follows:

- For method Gatys et al. (2016), we capture the vectorized features $F_\ell[\vec{x}]$ and $F_\ell[\vec{s}]$ of layers $\ell \in \{conv\mathbf{k}\_1\}_{\mathbf{k}=1}^5$ for the generated image $\vec{x}$ and the style image $\vec{s}$, and compute the Gram matrices:

$$G_\ell[\cdot] = F_\ell[\cdot]F_\ell[\cdot]^T, \tag{3}$$

here the Gram matrix is defined as the inner product between the vectorized features. In each layer, there are $N_\ell$ filters each with a vectorized feature of size $M_\ell$. We define our style loss $\mathcal{L}_1$ as follows:

$$\mathcal{L}_1 = \frac{1}{(2N_\ell M_\ell)^2}||G_\ell[\vec{x}] - G_\ell[\vec{s}]||^2. \tag{4}$$

- For method Champandard (2016) and Li & Wand (2016), we reuse the features $F_\hbar[\vec{x}]$ and $F_\hbar[\vec{s}]$ of layers $\hbar \in \{conv\mathbf{k}\_1\}_{\mathbf{k}=3}^4$, and then concatenate them with the segmentation masks $\vec{c}_m$ and $\vec{s}_m$ of the content image $\vec{c}$ and the style image $\vec{s}$ at the same resolutions, respectively:

$$\begin{aligned} F_{\hbar,m}[\vec{x}] &= F_\hbar[\vec{x}] \odot \lambda \cdot \eth[\vec{c}_m, \hbar], \\ F_{\hbar,m}[\vec{s}] &= F_\hbar[\vec{s}] \odot \lambda \cdot \eth[\vec{s}_m, \hbar], \end{aligned} \tag{5}$$

where $\eth[\cdot, \hbar]$ denotes resizing the masks to the same resolution as the output of layer $\hbar$. "$\odot$" denotes the channel concatenation, the hyperparameter $\lambda$ is given to weight the semantic awareness.

Then we extract a set of $3 \times 3$ neural patches for both $F_{\hbar,m}[\vec{x}]$ and $F_{\hbar,m}[\vec{s}]$, denoted by $\{\Phi_i(\vec{x})\}_{i \in n_x}$ and $\{\Phi_j(\vec{s})\}_{j \in n_s}$, where $n_x$ and $n_s$ are the number of extracted patches. For each patch $\Phi_i(\vec{x})$, we determine a closest-matching style patch $\Phi_{CM(i)}(\vec{s})$ based on the following measure:

$$CM(i) := \arg\max_{j=1,\ldots,n_s} \frac{\Phi_i(\vec{x}) \cdot \Phi_j(\vec{s})}{|\Phi_i(\vec{x})| \cdot |\Phi_j(\vec{s})|}. \tag{6}$$

Finally, our style loss $\mathcal{L}_2$ is defined as follows:

$$\mathcal{L}_2 = \sum_{i=1}^{n_x} ||\Phi_i(\vec{x}) - \Phi_{CM(i)}(\vec{s})||^2. \tag{7}$$

- For method Luan et al. (2017), we directly use their photorealism regularization term $\mathcal{L}_3$:

$$\mathcal{L}_3 = \sum_{c=1}^3 V_c[\vec{x}]^T \mathcal{M}_I V_c[\vec{x}], \tag{8}$$

where $\mathcal{M}_I$ is the Matting Laplacian Matrix Levin et al. (2008), which is used to express a locally affine combination of the input RGB channels and only depends on the input image $I$. $V_c[\vec{x}]$ denotes the vectorized version of the output image $\vec{x}$ in channel $c$.

• At last, these methods use the same content loss $\mathcal{L}_c$ to preserve the structure of the content image.

$$\mathcal{L}_c = ||\mathcal{F}[\vec{x}] - \mathcal{F}[\vec{c}]||^2, \tag{9}$$

where $\mathcal{F}[\vec{x}]$ and $\mathcal{F}[\vec{c}]$ denote the features extracted from layer $conv4\_2$.

**Total loss:** In our loss module, the total loss is the simple linear combination of the above loss:

$$\mathcal{L}_{total} = \alpha\mathcal{L}_c + \omega_1\mathcal{L}_1 + \omega_2\mathcal{L}_2 + \omega_3\mathcal{L}_3 + \mu\mathcal{L}_{TV}, \tag{10}$$

where $\mathcal{L}_{TV}$ refers to the total variation regularization loss Aly & Dubois (2005).

## 5 EXPERIMENTS

For SST, we select some of the most representative methods in specific domains, and run the published implementation with default stylization settings for each method.

For PST (we specifically refer to our *ParallelNet*), the hyperparameters $\alpha$, $\omega_1$, $\omega_2$ and $\mu$ are fixed at 10, 0.1, 10 and 1, respectively. $\lambda$ is set to 10 for cases with segmentation masks and 0 for others. $\omega_3$ is set to $10^4$ for photo-realistic style transfer, and 0 for others. The optimization is conducted by L-BFGS Zhu et al. (1997), and stopped at 500 iterations. The initialization is the content image.

### 5.1 ABLATION STUDIES

**How to select the appropriate methods?** To find some empirical principles, we conduct a user study (Table 1) on some representative NST methods in terms of content fidelity, stylization global color and local texture patterns, since these three aspects are our main considerations in selecting the methods. To alleviate the burden of subjects, we show them 150 synthesized images of each method, and ask them to select one overall level for each aspect. We collect totally 990 votes from 30 subjects, each method has 30 votes in each aspect. The levels with the most votes are chosen.

For **SST**, *which methods could be combined and which order will return the optimal solutions?* Take artistic style transfer as the example, Fig. 5 shows the comparisons of different serial schemes. The first column shows a reasonable serial scheme, since the method (b) help enrich the global color, and method (c) improves the local styles. The second column shows an unreasonable one, as the method (d) introduces some content distortions, and this directly affects the subsequent outputs. The third and fourth column show the effects of node order. We can find in the third column that method (b) and (a) progressively enrich the global color of (c), but do not change the local patterns. See the fourth column, unfortunately, changing the order of (d) cannot avoid the content distortions, but since the artistic style transfer does not require high content fidelity, the result is still acceptable. The last column shows some invalid nodes, method (b) and (e) have subtle effect on the result of (d). Now, look at these in conjunction with Table 1, we can conclude the following:

• For the aspects of global color and local patterns, the same level can promote each other, and the higher level can improve the performance of the lower level, while the lower level has little effect on the higher level. The results are determined by the order of methods with different levels.

• For the aspect of content fidelity, the higher level is only conducive to content preservation, while the distortions produced by the lower level are irreversible and sequence-independent.

• The selection is specific for particular domains, e.g., for artistic or semantic style transfer, the global color and local patterns are prior to content fidelity, while for photo-realistic style transfer, content fidelity is the primary concern.

These conclusions are also applicable to semantic and photo-realistic style transfer. More results can be found in appendix. We believe that these would help users design useful serial schemes to obtain desired results, and inspire future works in neural style transfer.

For **PST**, *how many and which methods could be combined?* As we introduced in Section 1, our PST is proposed to improve the quality and flexibility for different domains. Generally, any approach that satisfies the following two conditions can be incorporated into our PST:

Table 1: User study on some representative NST methods in terms of the content fidelity, global color and local patterns. The performance has three levels: H (High), M (Medium), L (Low).

| method | content fidelity | global color | local patterns | method | content fidelity | global color | local patterns |
|---|---|---|---|---|---|---|---|
| Sheng et al. (2018) | M | M | M | Luan et al. (2017) | H | M | L |
| Li et al. (2019) | H | M | L | Li et al. (2018) | M | M | L |
| Li & Wand (2016) | M | L | H | Gatys et al. (2016) | M | H | L |
| Champandard (2016) | M | L | H | Huang & Belongie (2017) | M | M | L |
| Gu et al. (2018) | M | M | H | Li et al. (2017b) | L | H | L |
| Liao et al. (2017) | M | M | H | | | | |

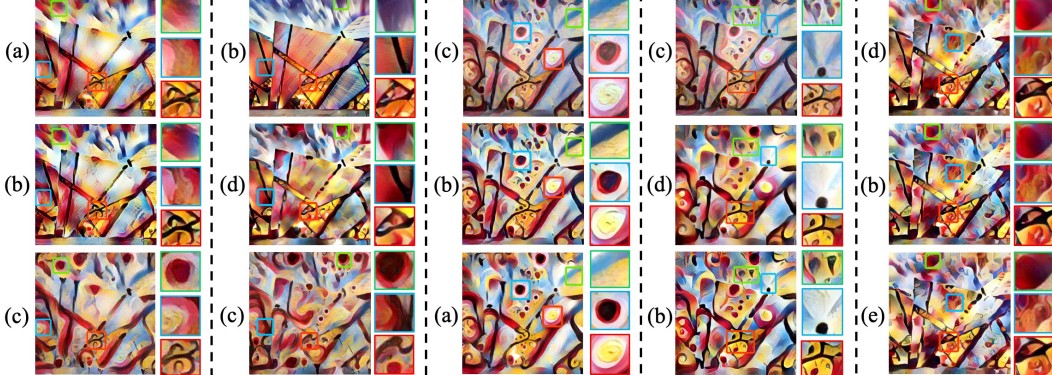

Figure 5: Comparisons of different serial schemes of some NST methods, i.e., (a) Sheng et al. (2018), (b) Li et al. (2019), (c) Li & Wand (2016), (d) Li et al. (2017b), (e) Huang & Belongie (2017). Each column shows the intermediate outputs and corresponding close-ups of a serial scheme.

• Sharing the same backbone (e.g., VGG-19) and similar process flow (e.g., backward optimization).

• Having the capacity to solve the problem of different style transfer domain.

**How to get the optimal hyperparameters for PST?** Our PST introduces some hyperparameters, we can adapt the model to different domains by adjusting them, but the fine-tuning may be in-tractable. Fortunately, we find that for most methods, the optimal hyperparameters before and after combinations are almost the same, e.g., the optimal values of $\lambda$ and $\omega_3$ of our *ParallelNet* are the same as those of the original methods. And for others, we just need to make a few adjustments according to the original methods. The optimal hyperparameters are generalized for most cases, but for some special ones, fine-tuning may produce better results. Ablation studies about different loss terms and their hyperparameters of our *ParallelNet* can be found in appendix.

## 5.2 EVALUATION AND COMPARISONS

**Qualitative Comparisons:** Fig. 6 shows some qualitative comparisons on artistic, semantic and photo-realistic style transfer. In the top row, we select Sheng et al. (2018), Li et al. (2019) and Li & Wand (2016) as three sequential nodes of our SST(a). Compared with them that only transfer the global color or local textures, our SST(a) and PST can consider both aspects simultaneously. In the middle row, we select Champandard (2016), Gu et al. (2018) and Liao et al. (2017) as three sequential nodes of our SST(b). We observe that these methods often produce either insufficiently stylized results or abnormal artifacts. By contrast, our SST(b) and PST produce much more satisfy-ing results. In the bottom row, since the content fidelity level of Li et al. (2018) in Table 1 is only M (we can also observe some undesired shadows in column 3) and this is detrimental to our SST in such high fidelity task, we only select Li et al. (2019) and Luan et al. (2017) as sequential nodes of our SST(c). As we can see, the result of method Li et al. (2019) may lose some striking styles, e.g.,

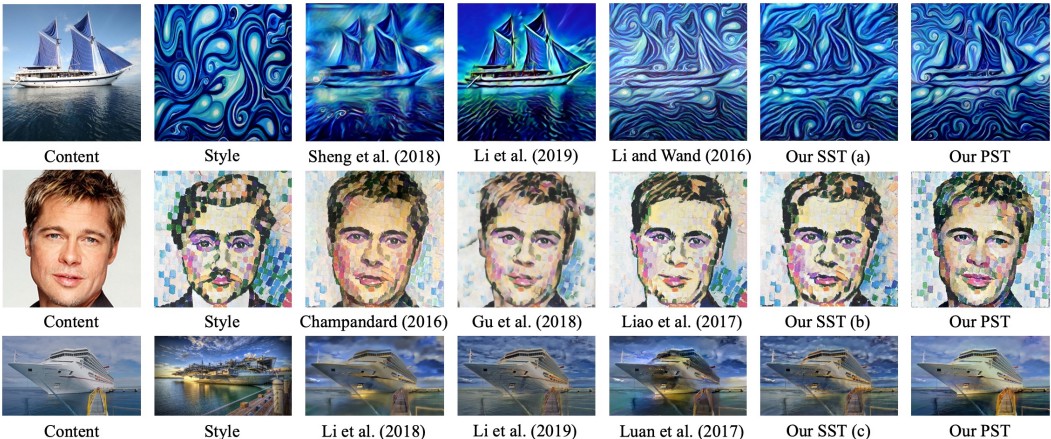

| Content | Style | Sheng et al. (2018) | Li et al. (2019) | Li and Wand (2016) | Our SST (a) | Our PST |
| Content | Style | Champandard (2016) | Gu et al. (2018) | Liao et al. (2017) | Our SST (b) | Our PST |
| Content | Style | Li et al. (2018) | Li et al. (2019) | Luan et al. (2017) | Our SST (c) | Our PST |

Figure 6: Qualitative comparisons in artistic (top), semantic (middle) and photo-realistic (bottom) style transfer. More results can be found in appendix.

Table 2: Flexibility evaluation of different NST methods.

| method | artistic | semantic (no mask) | semantic (mask) | photo-realistic | method | artistic | semantic (no mask) | semantic (mask) | photo-realistic |
|---|---|---|---|---|---|---|---|---|---|
| Sheng et al. (2018) | √ | √ | × | × | Gatys et al. (2016) | √ | × | × | × |
| Li et al. (2019) | √ | × | √ | √ | Huang & Belongie (2017) | √ | × | × | × |
| Li & Wand (2016) | √ | √ | × | × | Li et al. (2017b) | √ | × | × | × |
| Champandard (2016) | √ | √ | √ | × | Our SST(a) | √ | √ | × | × |
| Gu et al. (2018) | √ | √ | × | × | Our SST(b) | √ | √ | √ | × |
| Liao et al. (2017) | √ | √ | × | √ | Our SST(c) | × | × | √ | √ |
| Luan et al. (2017) | × | × | √ | √ | **Our PST** | √ | √ | √ | √ |
| Li et al. (2018) | × | × | √ | √ | | | | | |

the light at the bottom of the ship. The method Luan et al. (2017) may introduce some unacceptable artifacts, e.g., the abnormal hull. These problems do not occur in the results of our SST(c) and PST.

**Quantitative Comparisons:** We conduct several user studies on different style transfer domains. The results can be found in appendix. In all domains, our SSTs and PST show their superiority. And we also compare the flexibility of different methods in different style transfer domains in Table 2. Since our SSTs target the higher quality for specific domains, they do not contribute to more flexibility. By contrast, our PST is the only one that can flexibly adapt to all the mentioned domains.

**Speed and Memory Analysis:** Obviously, the time and memory requirements of our SSTs are simply the sums of those of its nodes. For PST, since the combined methods all share the same backbone, and the intermediate features can be stored and reused, the increments of time and memory are slight. The speed of our *ParallelNet* is comparable to that of the method Gatys et al. (2016).

## 6 CONCLUSION AND FUTURE DIRECTIONS

In this paper, we propose cascade style transfer to combine multiple approaches in serial or in parallel without modifying any algorithm. Experiments have verified that our methods can improve the stylistic quality and flexibility over previous state-of-the-art methods in artistic, semantic and photo-realistic style transfer domains. In the future, more effective and efficient schemes could be designed, and the combination of SST and PST is also an interesting direction worthy of further studies. Moreover, our methods can be regarded as data-flow graphs of neural image editing operators from a high-level perspective, it is another neat idea for interactive computer graphics system and will probably have considerable impact on future work and architectures in this domain.

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

# A APPENDIX

## A.1 MORE RESULTS ABOUT INITIALIZATION

### A.1.1 GATYS ET AL. (2016)

Fig. 7 shows more results of different initialization strategies on method Gatys et al. (2016). As mentioned in the paper, the first column shows the content image (top) and the style image (bottom). The other columns show the initialization images (top) and the corresponding style transfer results (bottom) of method Gatys et al. (2016). In column (f), we initialize with the default stylized results of method Li & Wand (2016) (SROOM). In column (g), we replace the content image with SROOM and then initialize with it (RC-SROOM).

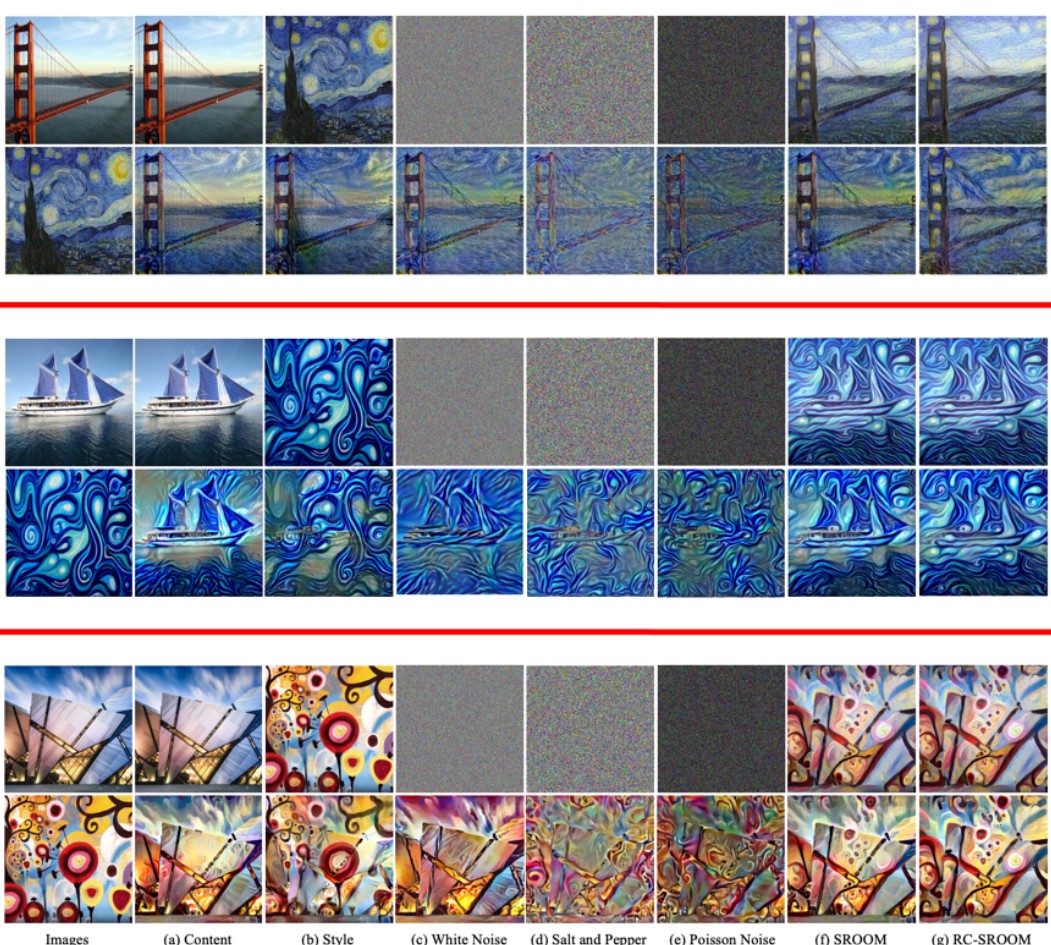

Figure 7: More results of different initialization strategies on method Gatys et al. (2016)

### A.1.2 LI & WAND (2016)

Fig. 8 shows more results of different initialization strategies on method Li & Wand (2016). The first column shows the content image (top) and the style image (bottom). The other columns show the initialization images (top) and the corresponding style transfer results (bottom) of method Li & Wand (2016). In column (f), we initialize with the default stylized results of method Gatys et al. (2016) (SROOM). In column (g), we replace the content image with SROOM and then initialize with it (RC-SROOM).

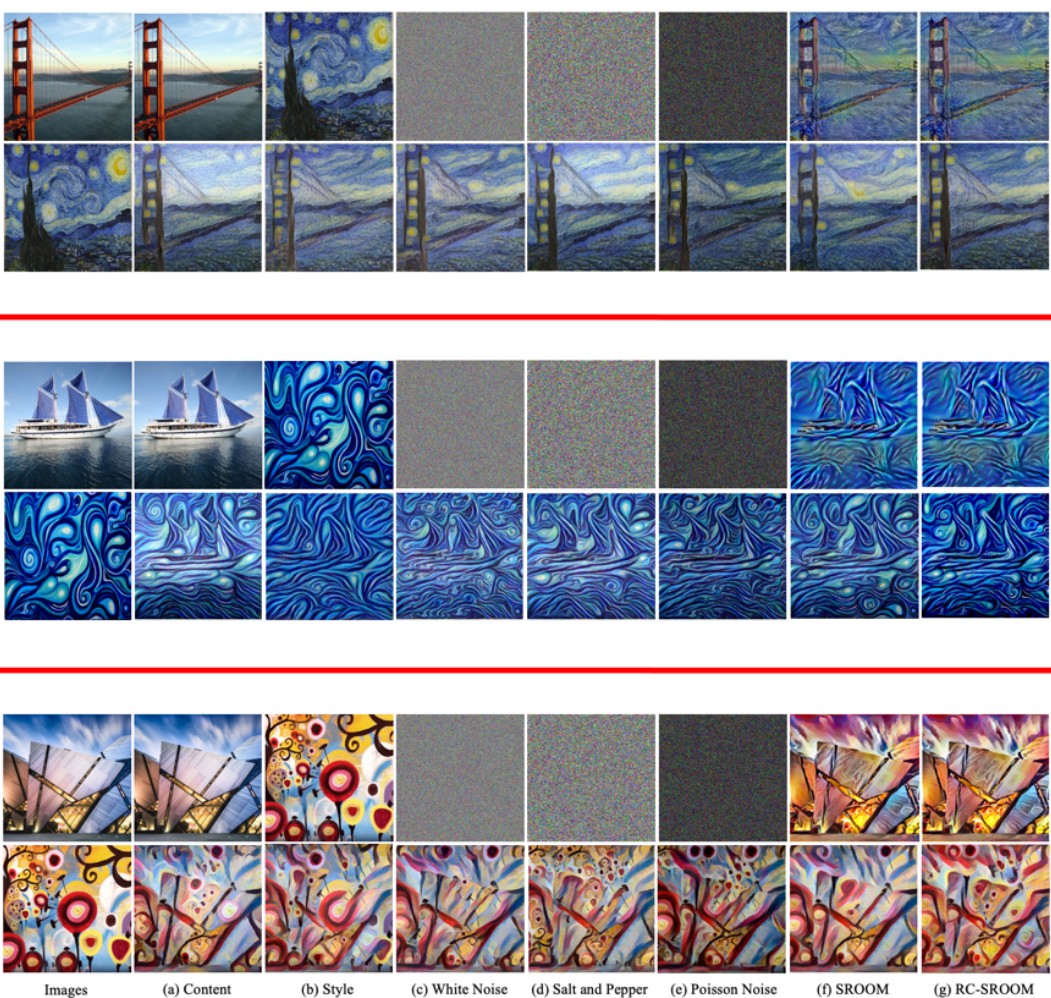

Figure 8: More results of different initialization strategies on method Li & Wand (2016)

## A.2 More Serial Schemes for SST

### A.2.1 Artistic Style Transfer

Fig. 9 shows more comparisons of different serial schemes in artistic style transfer domain. As mentioned in the paper, each column represents a serial scheme, and each row shows the intermediate output. We select (a) Sheng et al. (2018), (b) Li et al. (2019), (c) Li & Wand (2016), (d) Li et al. (2017b), (e) Huang & Belongie (2017) for the nodes of different serial schemes.

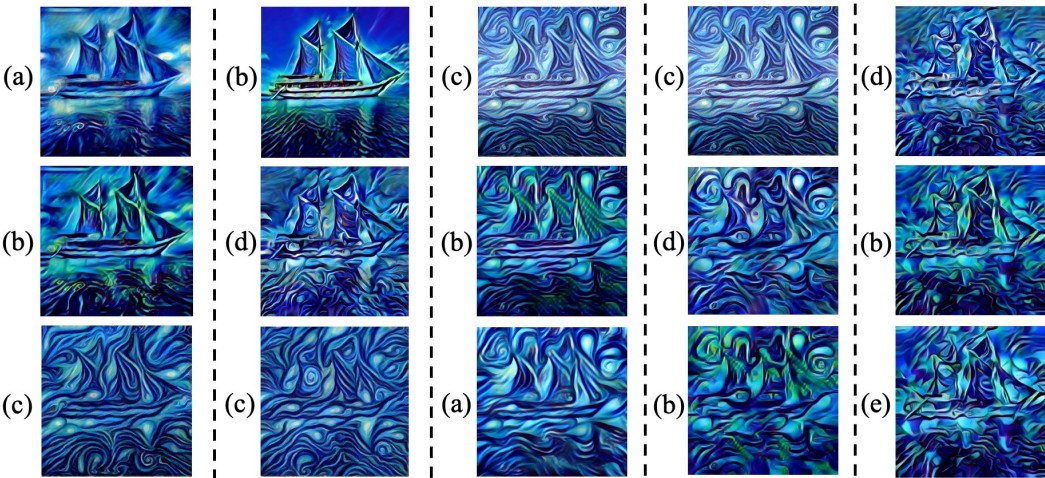

Figure 9: Comparisons of different serial schemes in artistic style transfer domain.

### A.2.2 Semantic Style Transfer

Fig. 10 shows some comparisons of different serial schemes in semantic style transfer domain. Each column represents a serial scheme, and each row shows the intermediate output. We select (a) Champandard (2016), (b) Gu et al. (2018), (c) Liao et al. (2017), (d) Li et al. (2017b), (e) Sheng et al. (2018), (f) Li et al. (2019) for the nodes of different serial schemes.

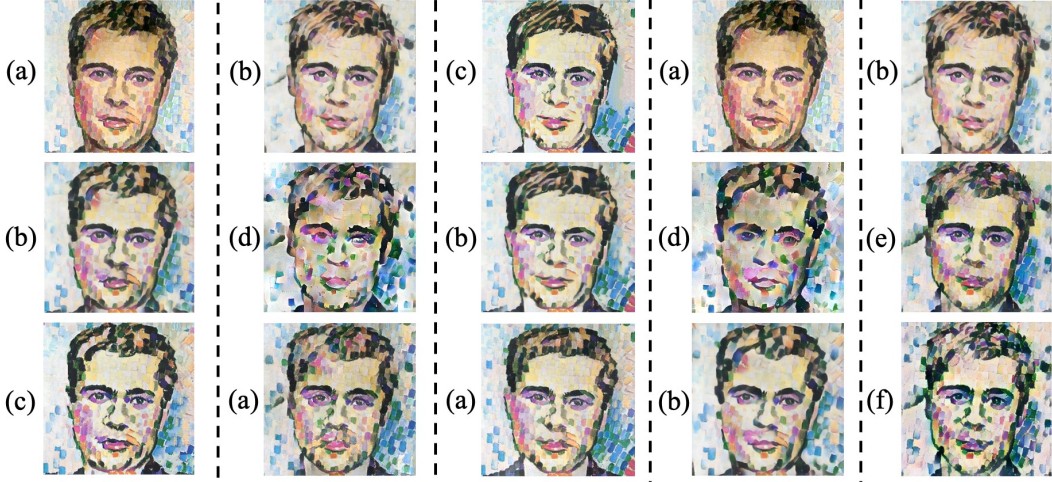

Figure 10: Comparisons of different serial schemes in semantic style transfer domain.

### A.2.3 PHOTO-REALISTIC STYLE TRANSFER

Fig. 11 shows some comparisons of different serial schemes in photo-realistic style transfer domain. Each column represents a serial scheme, and each row shows the intermediate output. We select (a) Li et al. (2019), (b) Luan et al. (2017), (c) Li et al. (2018), (d) Liao et al. (2017), (e) Huang & Belongie (2017) for the nodes of different serial schemes.

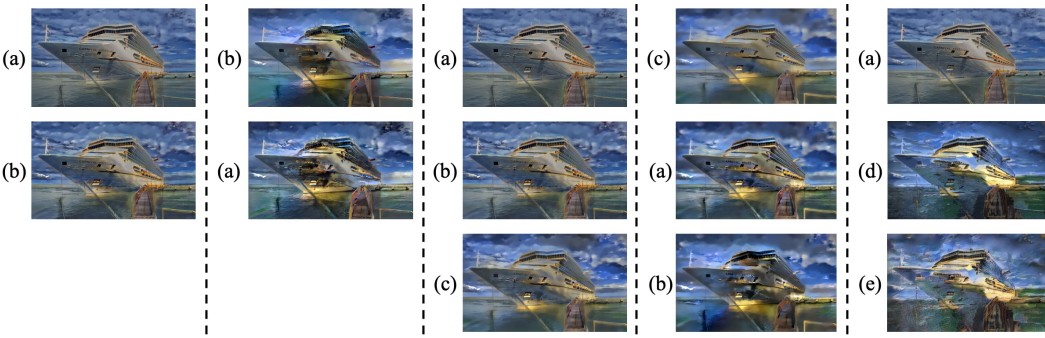

Figure 11: Comparisons of different serial schemes in photo-realistic style transfer domain.

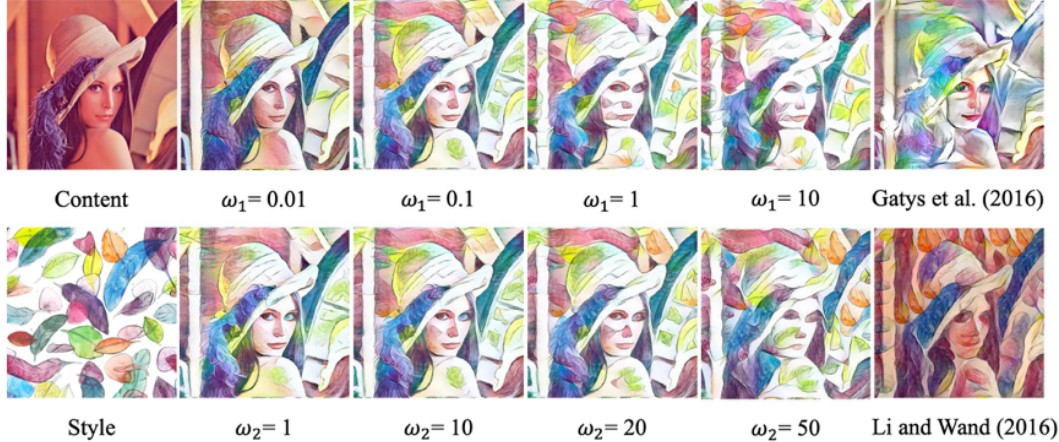

Figure 12: Varying weight $\omega_1$ of loss $\mathcal{L}_1$ and $\omega_2$ of loss $\mathcal{L}_2$ for artistic style transfer. The top row shows the results obtained by fixing $\omega_2$, $\omega_3$ and $\lambda$ at 10, 0 and 0, respectively, and increasing $\omega_1$ from 0.01 to 10. The bottom row shows the results obtained by fixing $\omega_1$, $\omega_3$ and $\lambda$ at 0.1, 0 and 0, respectively, and increasing $\omega_2$ from 1 to 50.

### A.3 ABLATION STUDIES OF PARALLELNET

We study the effects of different loss terms of our proposed *ParallelNet*, including style loss $\mathcal{L}_1$ (in Eq.(4)), $\mathcal{L}_2$ (in Eq.(5) and Eq.(7)) and photorealism regularization loss $\mathcal{L}_3$ (in Eq.(8)). Generally, the weight $\alpha$ of $\mathcal{L}_c$ is fixed to 10, and the weight $\mu$ of $\mathcal{L}_{TV}$ is fixed to 1. Fig. 12 shows the results of artistic style transfer by varying weight $\omega_1$ of loss $\mathcal{L}_1$ and $\omega_2$ of loss $\mathcal{L}_2$ while fixing the other weights. As the last column shows, the original method of Gatys et al. (2016) transfers the global color and rough textures of the style image, but does not transfer the local intricate patterns. The original method of Li & Wand (2016) transfers much more local style patterns, but as it shows, this method may generate insufficiently stylized result when huge difference exists between the content and style image. By combining the characteristics of these two methods, our *ParallelNet* can transfer both global color and local textures of the style images. As the top row shows, increasing the weight $\omega_1$ of loss $\mathcal{L}_1$ transfers more global color and rough textures. And as the bottom row shows, increasing the weight $\omega_2$ of loss $\mathcal{L}_2$ retains more local style patterns. However, since the trade-off

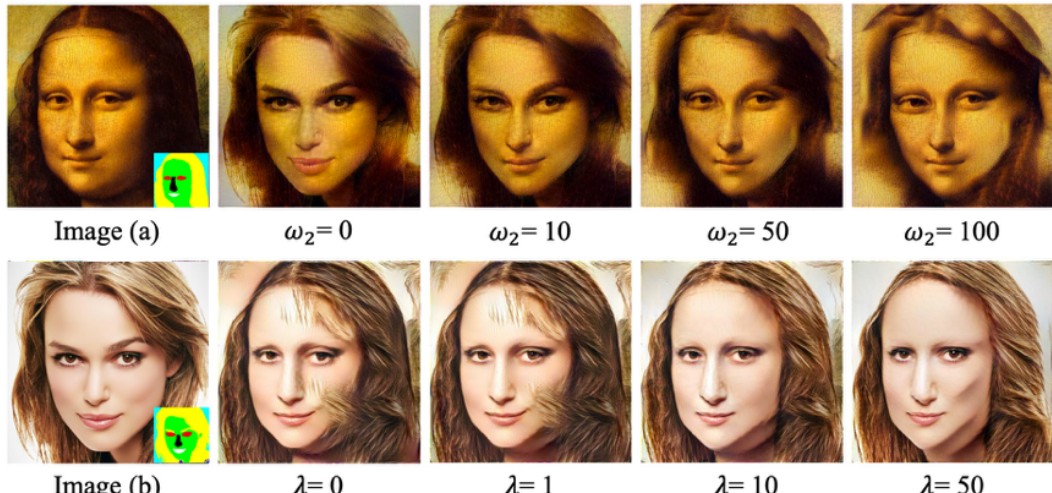

Figure 13: Varying weight $\omega_2$ and $\lambda$ of loss $\mathcal{L}_2$ for semantic style transfer. The top row shows the results obtained by fixing $\omega_1$, $\omega_3$ and $\lambda$ at 0.1, 0 and 0, respectively, and increasing $\omega_2$ from 0 to 100. The bottom row shows the results obtained by fixing $\omega_1$, $\omega_2$ and $\omega_3$ at 0.1, 10 and 0, respectively, and increasing $\lambda$ from 0 to 50.

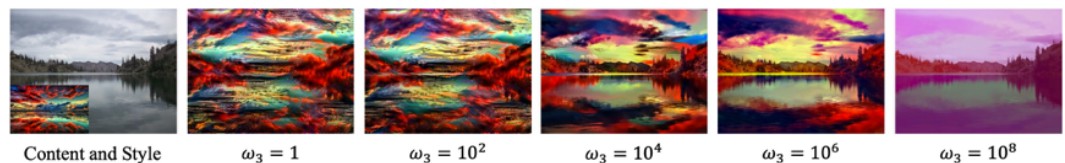

Figure 14: Varying weight $\omega_3$ of loss $\mathcal{L}_3$ for photo-realistic style transfer. The values of $\omega_1$, $\omega_2$ and $\lambda$ are fixed at 0.1, 10 and 10, respectively.

between the content loss and the style loss, distortions of the content structure are inevitable if the values of $\omega_1$ and $\omega_2$ are too high. In our work, for artistic style transfer, $\omega_1$ and $\omega_2$ are set to 0.1 and 10 by default, respectively.

Fig. 13 shows the results of semantic style transfer by varying weight $\omega_2$ and $\lambda$ of loss $\mathcal{L}_2$. The top row shows transfer of painted style (a) onto a photo (b), this is easy and does not require the constraints of segmentation masks, so we fix the semantic awareness weight $\lambda$ at 0. On the other hand, according to the aforementioned practice, we set the value of $\omega_1$ and $\omega_3$ to 0.1 and 0 by default, respectively. As the top row shows, increasing $\omega_2$ refines more detailed information of the corresponding semantics, but this may change the original structure of the content image. To avoid this, the value of $\omega_2$ should not be too high, so in our work, we set it to 10 by default. Based on these, the bottom row shows transfer of photo style (b) onto a painting (a), it is hard so we use the constraints of segmentation masks to get better results. As the bottom row shows, increasing $\lambda$ achieves more accurate semantic matching. To obtain the best results, we set $\lambda$ to 10 by default.

Fig. 14 shows the results of photo-realistic style transfer by varying weight $\omega_3$ of photorealism regularization loss $\mathcal{L}_3$. Differing from Luan et al. (2017) using a two-stage optimization process based on the outputs of Gatys et al. (2016), we directly solve this problem by optimizing Eq. (10). This is much simpler, and could produce comparable results due to the synergistic effects of different loss items. As we can see, a too small value of $\omega_3$ cannot prevent distortions, thus the results in column 2 and 3 have a non-photorealistic look. On the contrary, a too large value of $\omega_3$ suppresses the style to be transferred and leads to color infidelity (see column 5 and 6). Similar to Luan et al. (2017), we choose $\omega_3 = 10^4$ by default.

### A.4 QUALITATIVE COMPARISONS WITH OTHER METHODS

Here we demonstrate the results of our methods and twelve other representative single- or multi-domain methods Sheng et al. (2018); Li et al. (2019); Li & Wand (2016); Champandard (2016); Gu et al. (2018); Liao et al. (2017); Luan et al. (2017); Li et al. (2018); Yao et al. (2019); Gatys et al. (2016); Huang & Belongie (2017); Li et al. (2017b) on artistic, semantic and photo-realistic style transfer, respectively. Each figure shows the representative results obtained by one method. The left, center and right columns show examples of artistic, semantic and photo-realistic style transfer, respectively. For the inputs in each group, the upper one is the content image and the lower one is the style image. Relevant quality and flexibility analyses can be found under the caption of each figure. Compared to these methods, our SST schemes perform better on stylistic quality in specific domains, and our PST scheme performs better on stylistic quality and flexibility in all domains.

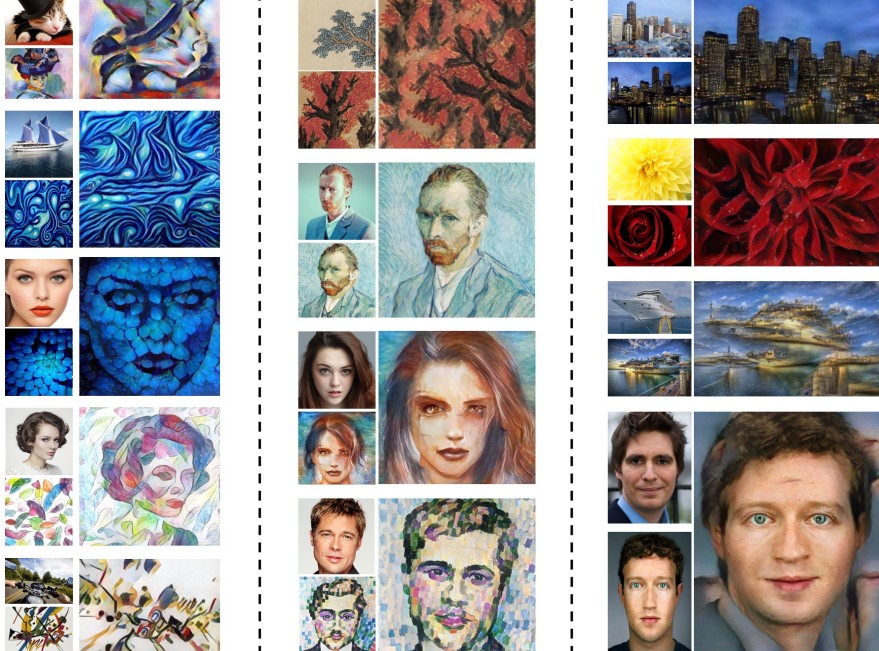

Figure 15: Results obtained by our SST(a) scheme.

As shown in Fig. 15, this scheme can perform well on artistic and semantic style transfer (without segmentation masks). But since its nodes do not support segmentation masks, it could not solve the cases when huge differences exist between the content and style images (e.g., the case at the top of the center column). This scheme is not suitable for photo-realistic style transfer, as it may produce a lot of content distortions and abnormal artifacts (see the right column).

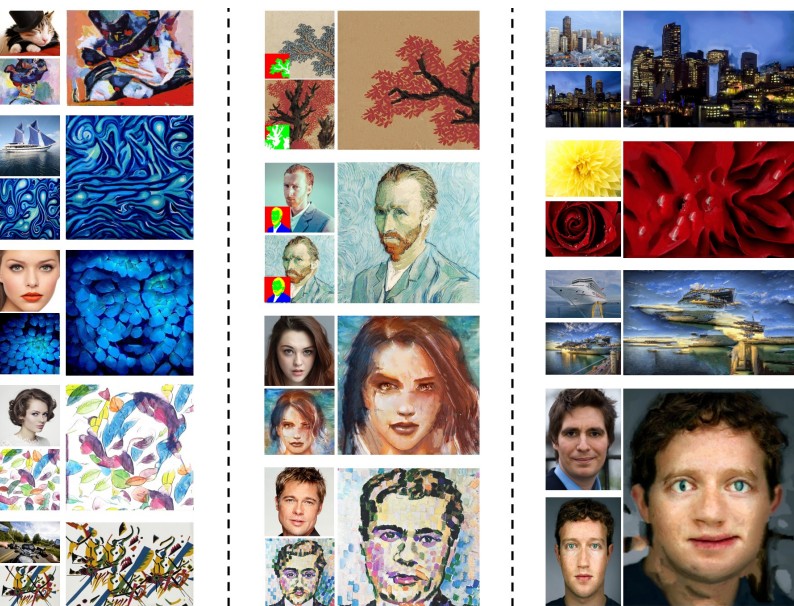

Figure 16: Results obtained by our SST(b) scheme.

As shown in Fig. 16, this scheme can perform well on artistic and semantic style transfer (with and without segmentation masks). However, in artistic style transfer, there are some deficiencies in content preservation because it prefers to express more style features (see the two cases at the bottom of the left column). This also limits its capability in photo-realistic style transfer (see the right column).

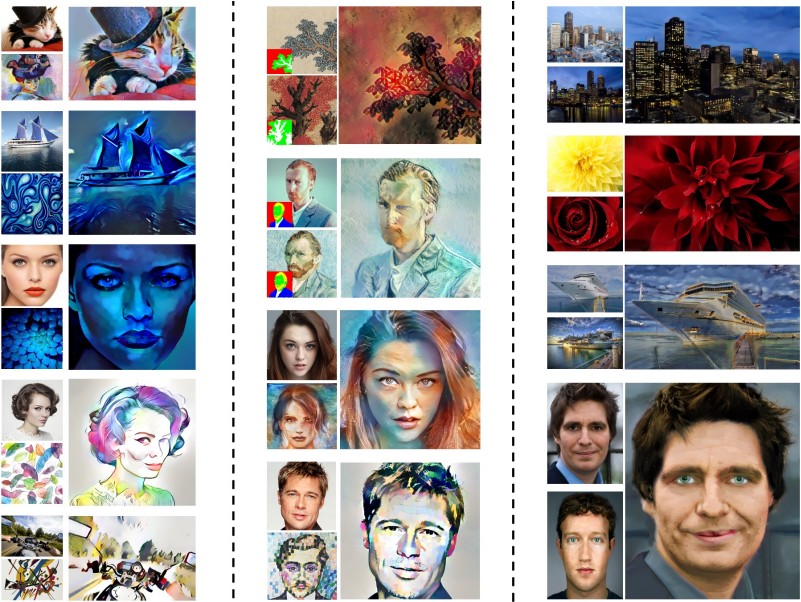

Figure 17: Results obtained by our SST(c) scheme.

As shown in Fig. 17, this scheme is only suitable for photo-realistic style transfer. As we can see, it maintains the photorealism of the content photographs and at the same time transfers the global color of the style images. But this also makes it difficult to produce the artworks with non-realistic styles. Since the nodes of it can incorporate segmentation masks, this scheme is capable to use segmentation masks.

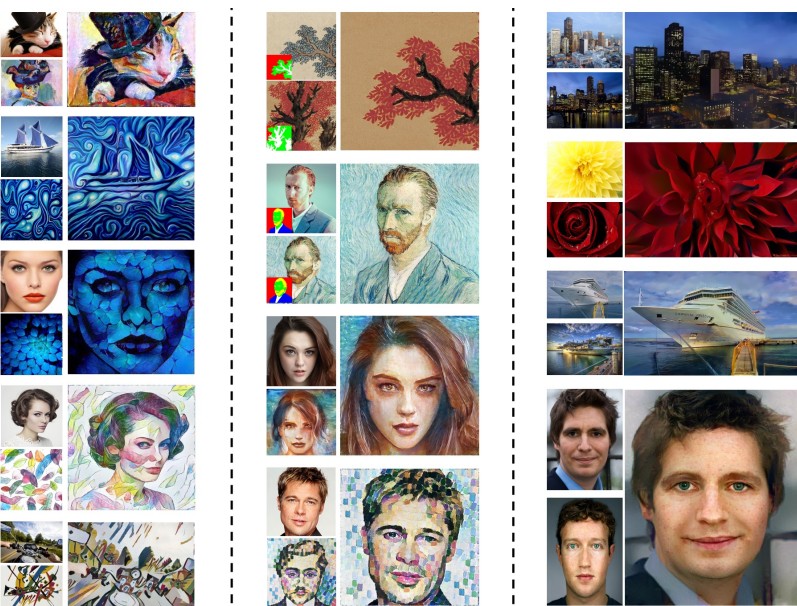

Figure 18: Results obtained by our PST scheme.

As shown in Fig. 18, this scheme is flexible enough to apply to all these style transfer domains. As we can see, in the left column, both the global color and local patterns of the artistic images can be transferred by this scheme. In the center and right column, it can also achieve semantic and photo-realistic style transfer with and without segmentation masks. Moreover, users can further improve the stylistic quality by fine-tuning our provided hyperparameters for each input.

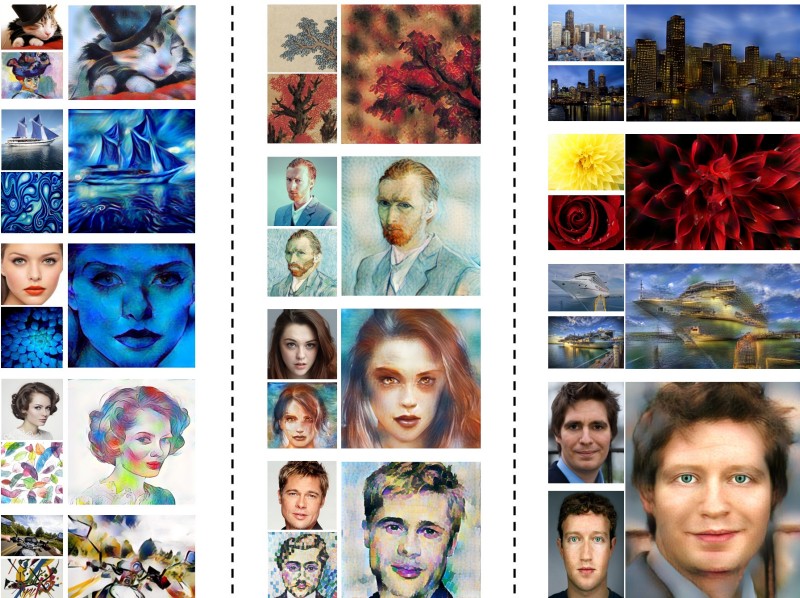

Figure 19: Results obtained by the method of Sheng et al. (2018)

As shown in Fig. 19, this method can be used in artistic and semantic style transfer (without segmentation masks), but the stylistic quality is not so good. As we can see, for artistic style transfer, it only transfers the global color of the style images, but lacks local patterns. For semantic style transfer, it may produce a lot of hazy blocks, which affect the overall clarity. These problems (including content distortions and abnormal artifacts) also exist in the photo-realistic style transfer.

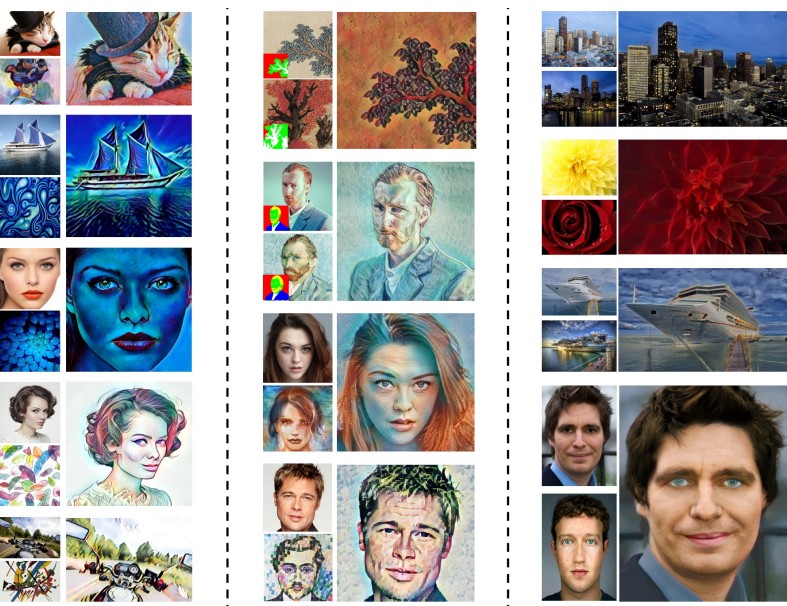

Figure 20: Results obtained by the method of Li et al. (2019)

As shown in Fig. 20, this method can be used in artistic and photo-realistic style transfer. But since it prefers to maintain the structures of the content images, the produced results are often insufficiently stylized. This could help it to perform better in photo-realistic style transfer. On the other hand, since the spirit of this method is based on the global statistics, it cannot solve the tasks of semantic style transfer. This method is capable to use segmentation masks.

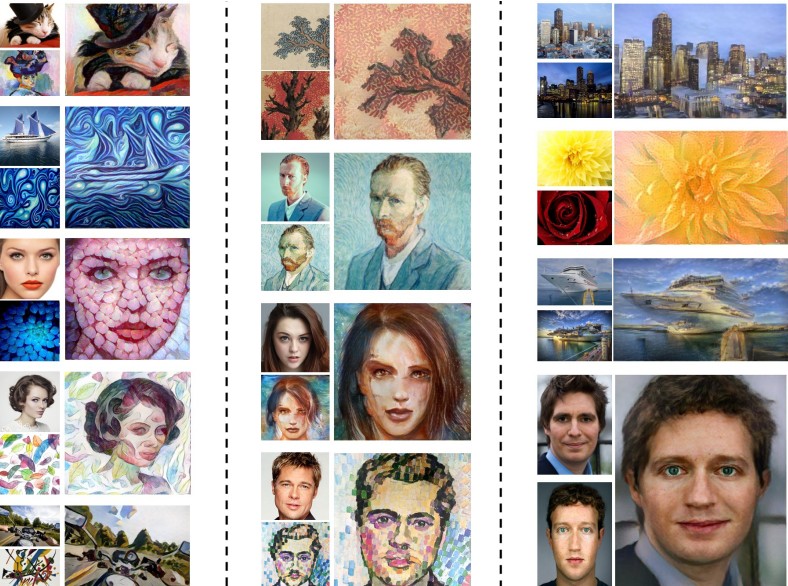

Figure 21: Results obtained by the method of Li & Wand (2016)

As shown in Fig. 21, this method can be used in semantic style transfer (without segmentation masks). As we can see, for artistic style transfer, it could transfer adequate local patterns of the style images, but the global effects of the stylized results are unsatisfying (mainly because of the insufficient global color, see the left column). On the other hand, introduced content distortions also limit its capability for photo-realistic style transfer (see the right column).

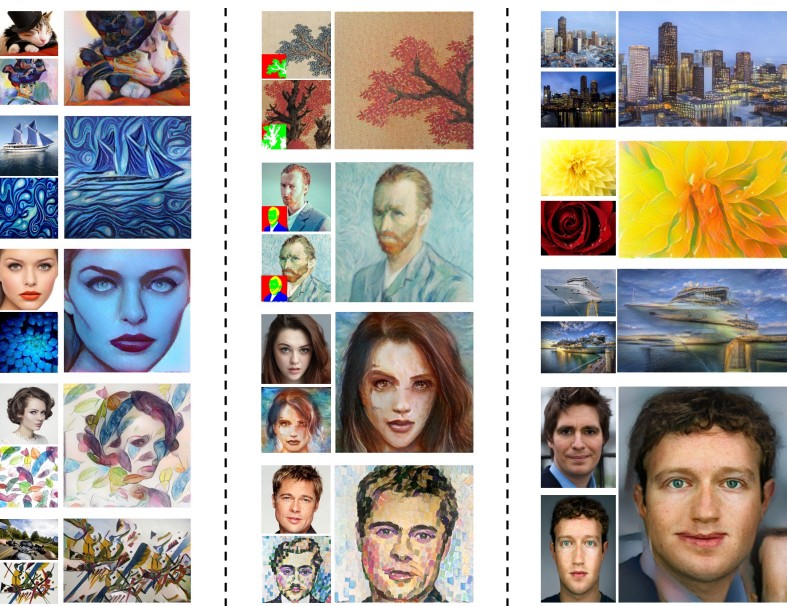

Figure 22: Results obtained by the method of Champandard (2016)

As shown in Fig. 22, this method can be used in artistic and semantic style transfer (with and without segmentation masks). For artistic style transfer (see the left column), the most results are similar to that of Li & Wand (2016), the difference is that this method can perform better on the global color or the local patterns, but it is still unsatisfying. For semantic style transfer, it may produce blurred results (e.g., the two cases at the top of the center column). For photo-realistic style transfer, it cannot adapt to this domain because of the introduced content distortions.

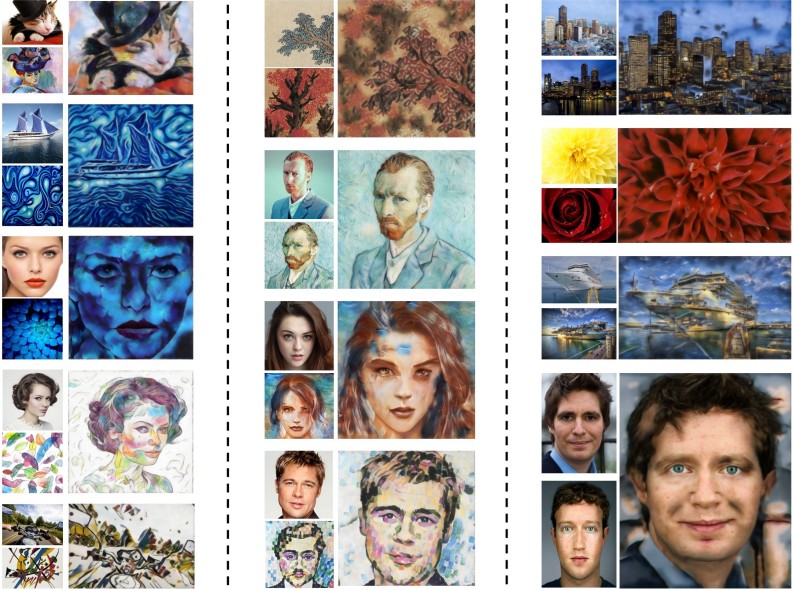

Figure 23: Results obtained by the method of Gu et al. (2018)

As shown in Fig. 23, this method can be used in artistic and semantic style transfer (without segmentation masks). The stylized results produced by it can obtain sufficient global color and local patterns, but there are also a lot of abnormal artifacts which directly affect the final effect. This occurs in almost every case in these three domains.

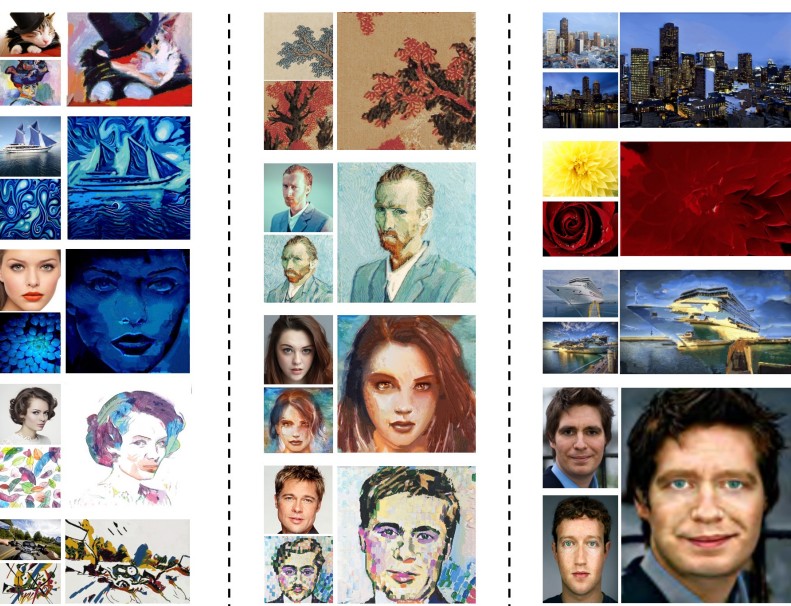

Figure 24: Results obtained by the method of Liao et al. (2017)

As shown in Fig. 24, this method can be used in semantic and photo-realistic style transfer (without segmentation masks). It is more suitable for style transfer of image pairs which have high semantic-level correspondences. Therefore, for the cases of artistic style transfer which are not relevant in semantics, this method yields poor stylized results (see the left column). On the other hand, since this method does not support segmentation masks, mismatching is prone to occur in challenging tasks (e.g., the case at the top of the center column and cases in the right column).

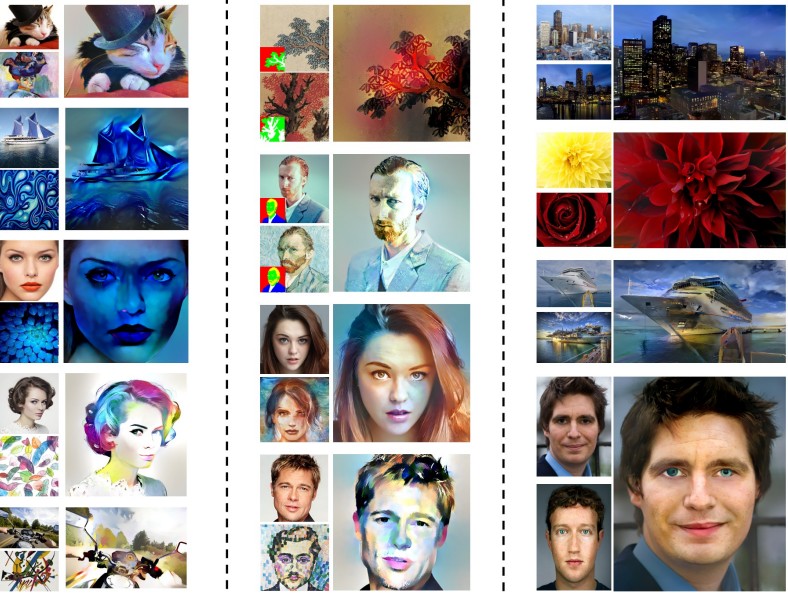

Figure 25: Results obtained by the method of Luan et al. (2017)

As shown in Fig. 25, similar to our SST(c) scheme (see Fig. 17), this method is only suitable for photo-realistic style transfer. Compared to our SST(c) scheme, it may produce some abnormal artifacts, e.g., the hull and eyes in the two cases at the bottom of the right column, respectively. This method is capable to use segmentation masks.

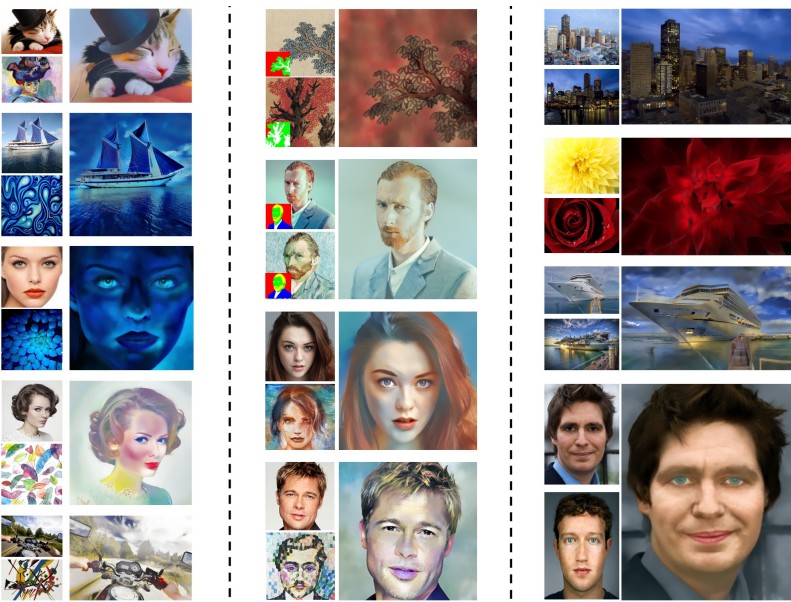

Figure 26: Results obtained by the method of Li et al. (2018)

As shown in Fig. 26, similar to our SST(c) scheme (see Fig. 17) and the method of Luan et al. (2017) (see Fig. 25), this method is only suitable for photo-realistic style transfer. See the right column, compared to our SST(c) scheme, it may produce too many undesired effects, e.g., the skies in the case 1 and case 3, the shadows in the case 2 and the eyes in the case 4 (numbers are arranged from top to bottom). This method is capable to use segmentation masks.

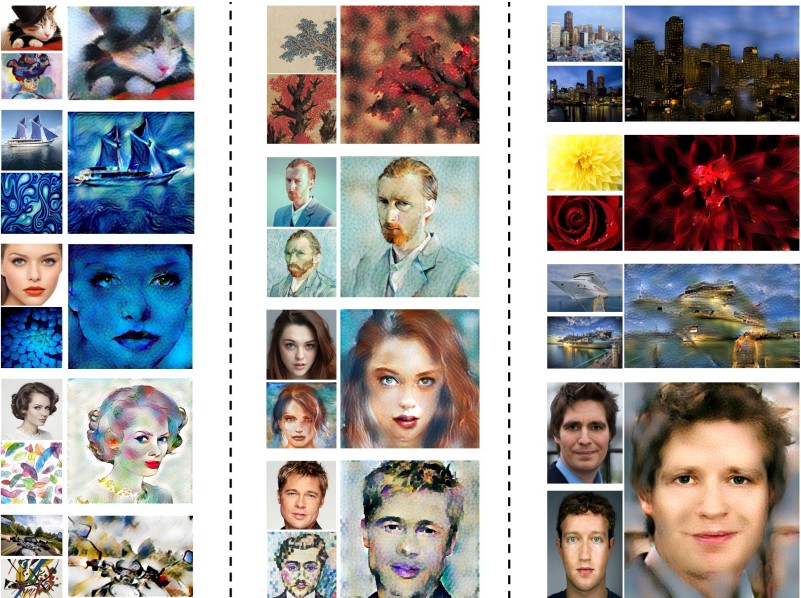

Figure 27: Results obtained by the method of Yao et al. (2019)

As shown in Fig. 27, this method can be used in artistic and semantic style transfer (without segmentation masks). Because of the incorporation with self-attention mechanism, this method can highlight more salient areas (e.g., characters' eyes) of the images. But in other places, the performance is similar to the method of Sheng et al. (2018) (see Fig. 19).

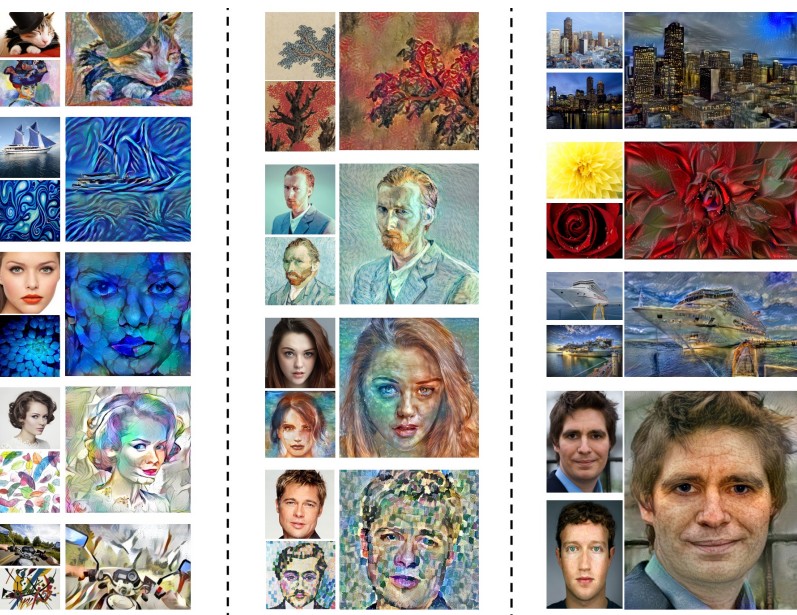

Figure 28: Results obtained by the method of Gatys et al. (2016)

As shown in Fig. 28, this method is only suitable for artistic style transfer. As we can see, it could transfer the global color and rough textures of the artistic style images to the content images (see the left column). But since the spirt of its algorithm is based on the global statistics, this method cannot solve the semantic style transfer (see the center column). Of course, without the improvement introduced by Luan et al. (2017), it cannot solve the tasks of photo-realistic style transfer (see the right column).

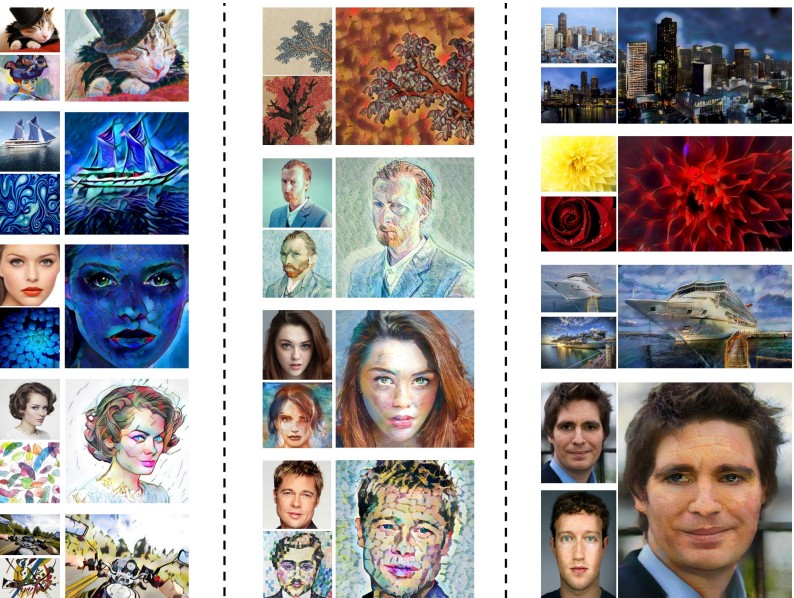

Figure 29: Results obtained by the method of Huang & Belongie (2017)

As shown in Fig. 29, similar to the method of Gatys et al. (2016) (see Fig. 28), this method is only suitable for artistic style transfer. Compared to the method of Gatys et al. (2016), some results generated by it are not sufficiently stylized (e.g., the two cases at the bottom of the left column).

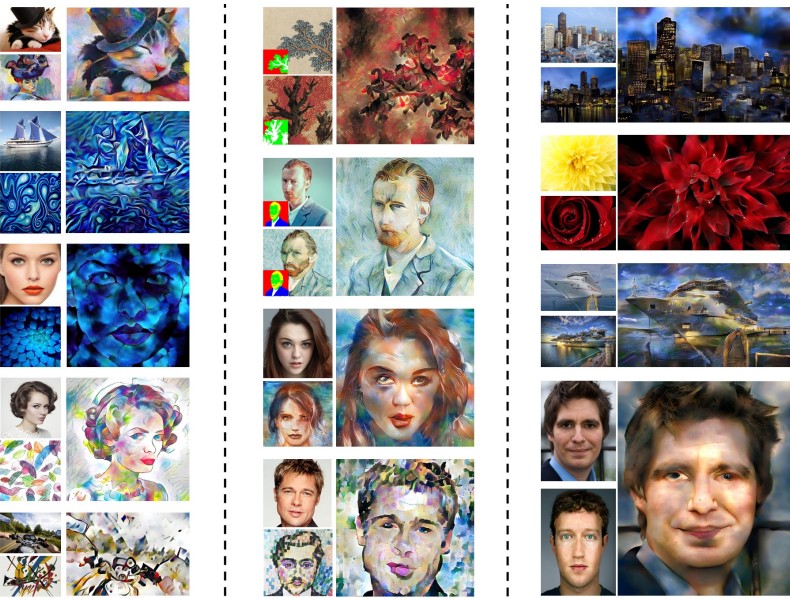

Figure 30: Results obtained by the method of Li et al. (2017b)

As shown in Fig. 30, similar to the methods of Gatys et al. (2016) (see Fig. 28) and Huang & Belongie (2017) (see Fig. 29), this method is only suitable for artistic style transfer. Compared to the methods of Gatys et al. (2016) and Huang & Belongie (2017), some results generated by it are excessively stylized, thus introducing a lot of undesired effects (e.g., the case 2 and case 3 in the left column).

## A.5 QUANTITATIVE COMPARISONS WITH OTHER METHODS

### A.5.1 USER STUDY ON ARTISTIC STYLE TRANSFER

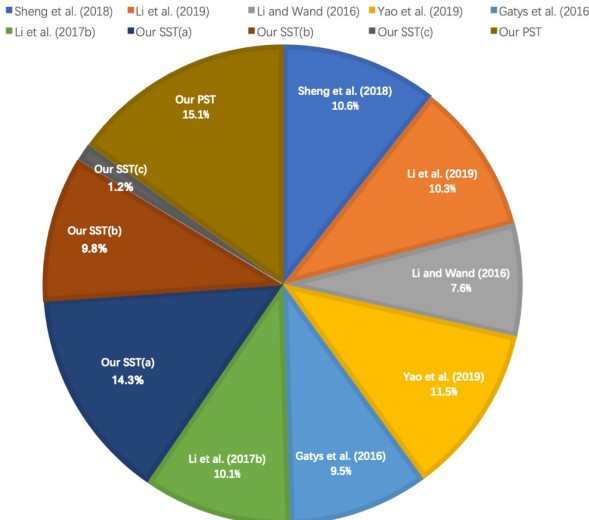

Figure 31: User preference result of ten style transfer methods for artistic style transfer.

We conduct a user study to evaluate the proposed schemes against the state-of-the-art style transfer methods Sheng et al. (2018); Li et al. (2019); Li & Wand (2016); Yao et al. (2019); Gatys et al. (2016); Li et al. (2017b) on artistic style transfer. We use 10 content images and 10 style im-

ages to synthesize 100 images in total for each method, and randomly select 50 content and style combinations to each subject. We show stylized images of 10 compared methods (including ours) side-by-side in a random order and ask the subjects to select the most visually pleasant one. We collect 1500 votes from 30 users and show the percentage of votes for each method in Fig. 31. Overall, our proposed PST and SST(a) are favored among all evaluated methods.

### A.5.2 USER STUDY ON SEMANTIC STYLE TRANSFER

We conduct a user study to evaluate the proposed schemes against the state-of-the-art style transfer methods Sheng et al. (2018); Li & Wand (2016); Champandard (2016); Gu et al. (2018); Liao et al. (2017); Yao et al. (2019) on semantic style transfer. For each method, we use 30 image groups to synthesize 30 images (all images are produced without using segmentation masks) in total. We show stylized images of 10 compared methods (including ours) side-by-side in a random order and ask the subjects to select the most visually pleasant one. We collect 900 votes from 30 users and show the percentage of votes for each method in Fig. 32 (a). Overall, our proposed PST, SST(b) and SST(a) are favored among all evaluated methods.

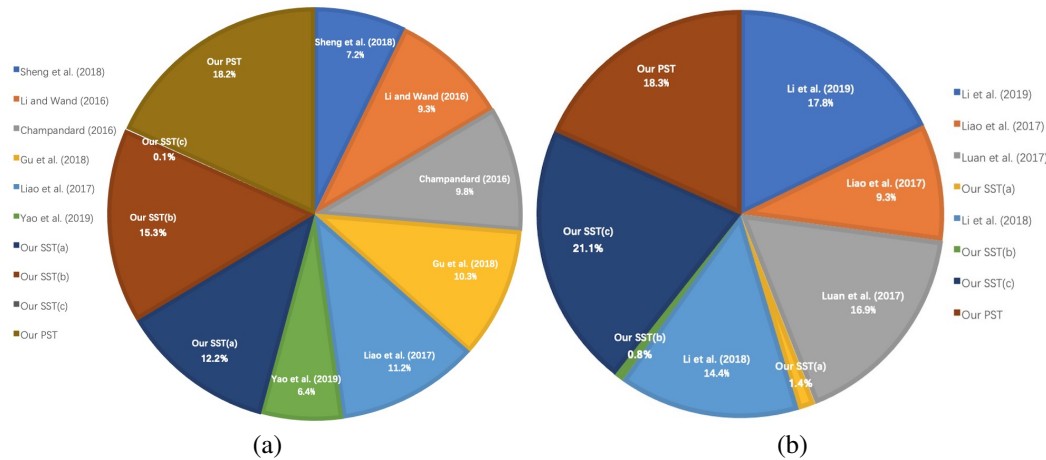

Figure 32: (a) User preference result of ten style transfer methods for semantic style transfer. (b) User preference result of eight style transfer methods for photo-realistic style transfer.

### A.5.3 USER STUDY ON PHOTO-REALISTIC STYLE TRANSFER

We conduct a user study to evaluate the proposed schemes against the state-of-the-art style transfer methods Li et al. (2019); Liao et al. (2017); Luan et al. (2017); Li et al. (2018) on photo-realistic style transfer. For each method, we use 30 image groups to synthesize 30 images in total. We show stylized images of 8 compared methods (including ours) side-by-side in a random order and ask the subjects to select the most visually pleasant one. We collect 900 votes from 30 users and show the percentage of votes for each method in Fig. 32 (b). Overall, our proposed SST(c) and PST are favored among all evaluated methods.

