# OpenReview forum: "Cascade Style Transfer"
_ICLR.cc/2020/Conference — Reject_

### Official Review · AnonReviewer1 · 2019-10-14
**Official Blind Review #1**

**Rating:** 1

**Review:**

Summary:
In this study, the authors propose a new method for performing artistic style transfer for  arbitrary image and styles. The new method employs a cascade/serial architecture for performing the style transfer. The authors test their method using human preference studies.

In summary, I found the architecture choice to be minimally explored. More importantly, a vast majority of the results to demonstrate the relative merits of this method were qualitative. The minimal quantitative results were unconvincing and left many unanswered questions about how well one could trust these results.

Major Comments:

1. No experiments to explore the architecture hyperparameters.
A natural question might be how the quality of the method varies systematically as the number of methods N grows. Presumably, if N=1, this would recover previous methods.

2. Authors are missing an important reference and point of comparison for arbitrary style transfer.
  Exploring the structure of a real-time, arbitrary neural artistic stylization network
  Golnaz Ghiasi, Honglak Lee, Manjunath Kudlur, Vincent Dumoulin, Jonathon Shlens
  https://arxiv.org/abs/1705.06830
  http://goo.gle/2oiDKaT

3. Minimal quantitative analysis.
A vast majority of the results (30 of 32 figures) are qualitative comparisons and the paper is sorely lacking an emphasis on quantitative comparisons. This is a large and notable problem in this paper and a quantitative comparison *should* constitute the primary thrust and central result of such a paper to convincingly demonstrate to a reader that the proposed method is indeed to superior to other techniques. I wish the authors dedicated more emphasis in this paper to a detailed quantitative comparison for these methods. As a starter, the analysis presented as the final two appendix figures (31 and 32) should be front and center in the result section of the paper.

4. User study for quantitative comparison is incomplete and unconvincing.
Table 1 and Appendix Figure 31 and 32 represent the primary result of this paper as these results and comprise the user studies to quantify how much better this method is to previous methods. These studies however are fairly unconvincing as lots of details are omitted and and I am concerned about the rigor of the human studies including but not limited to:
  4a. How long did each human study each image? What controls were added to the study to ensure that all images were equally studied by humans? For instance, were any golden tests employed to ensure user engagement throughout the study?
  4b. What was the repeatability of each measurement of preference? If a single human was presented the same image twice, how consistent were there ratings? For that matter, how consistent were the ratings across humans? I presume that some humans preferred some styles over others but how systematic was this?
  4c. What types of user testing scenarios were explored to ensure minimal bias in the results? Were multi-choice, paired choice or force choice employed? What about minimal or maximal time limit enforcement?
  4d. How can I have confidence that the authors did not cherry pick images and styles that favored their method? For that matter, I would expect that some methods work better on some styles or images. I would expect to see analysis accordingly to break down which styles/images work better on different slices of the data.
  4e. The statistical significance of Figure 31 and Figure 32 is not provided. What would an error bar look like with resampling?

**Experience Assessment:**

I have published one or two papers in this area.

**Review Assessment: Checking Correctness Of Derivations And Theory:**

I assessed the sensibility of the derivations and theory.

**Review Assessment: Checking Correctness Of Experiments:**

I carefully checked the experiments.

**Review Assessment: Thoroughness In Paper Reading:**

I read the paper thoroughly.

---

### Official Review · AnonReviewer2 · 2019-10-31
**Official Blind Review #2**

**Rating:** 1

**Review:**

The authors proposed to mix together multiple styles by proposing two frameworks: 1) serial style transfer (SST), which combines style transfer methods in series; and 2) parallel style transfer (PST), which combines style transfer methods in parallel.

The paper is clearly presented. It is interesting to see work on mixing up different styles, since it is not extensive studied so far. Though not much studied, this topic is not new [ref 1], [ref 2]. The authors didn't provide a thorough literature review on mixing multiple styles in the related work or anywhere else in the submission.

In terms of the methodology, the novelty is quite limited. The proposed SST and PST are simple frameworks to mix different styles, which, by the way, are fully based on existing style transfer methods. At some point, PST is similar to [ref 2], and the difference is minor. PST linearly combines the losses for different styles to construct the final loss, while [ref 2] linearly combines together the features for different styles.

About experiments, it is not clear about how the predefined parameters (e.g. \alpha, w_1, w_2, etc.) are determined. They were just empirically set and mentioned in the experiment section.

It is appreciated to see more results in the Appendix, as well as the user study. However, due to lack of novelty, I think this submission may not be qualified for acceptance at this moment.

Minor:
I think the authors should give their proposed framework another name instead of using "cascade," which has a similar meaning of "series."


[ref 1] Google's arty filters one-up Prisma by mixing various styles. https://www.engadget.com/2016/10/27/google-style-transfer-tech/
[ref 2] Pegios, et al. Style Decomposition for Improved Neural Style Transfer. ArXiv, 2018.

**Experience Assessment:**

I have read many papers in this area.

**Review Assessment: Checking Correctness Of Derivations And Theory:**

I carefully checked the derivations and theory.

**Review Assessment: Checking Correctness Of Experiments:**

I carefully checked the experiments.

**Review Assessment: Thoroughness In Paper Reading:**

I read the paper at least twice and used my best judgement in assessing the paper.

---

### Official Review · AnonReviewer4 · 2019-11-03
**Official Blind Review #4**

**Rating:** 1

**Review:**

This paper proposes two ways to aggregate existing style transfer methods and shows improvements on quality and flexibility. However, the proposed method does not solve the limitations of any previous methods. Instead, it is as simple as an easy combination: the proposed SST is a sequential combination and the proposed PST has no difference with running each single method separately. To me this is more like an engineering effort rather than a research work.

(1) For SST, it just connects N existing methods, using the output of method 1 as the input of method 2. The quality of results might be improved but there is little novelty. I agree combing methods can be a contribution only when there are principle designs and in-depth analysis.

(2) For PST, I do not see its difference with running single method separately. Putting all previous methods altogether cannot be called being more flexible. As said in the paper, when for photorealistic transfer, the proposed PST set the loss weight of other methods except Luan et al. as 0. Then is it the same with running the single method of Luan et al.?

In general, I do not encourage such a way of exploring research. Authors should focus more on the unsolved issues in style transfer, e.g., how to do geometric style transfer (shape).

**Experience Assessment:**

I have published in this field for several years.

**Review Assessment: Checking Correctness Of Derivations And Theory:**

I carefully checked the derivations and theory.

**Review Assessment: Checking Correctness Of Experiments:**

I carefully checked the experiments.

**Review Assessment: Thoroughness In Paper Reading:**

I read the paper thoroughly.

---

### Decision · Program_Chairs · 2019-12-19

**Decision:**

Reject

**Comment:**

This work combines style transfer approaches either in a serial or parallel fashion, and shows that the combination of methods is more powerful than isolated methods.
The novelty in this work is extremely limited and not offset by insightful analysis or very thorough experiments, given that most results are qualitative. Authors have not provided a public response.
Therefore, we recommend rejection.